# Determinants of the Essential Elements and Vitamins Intake and Status during Pregnancy: A Descriptive Study in Polish Mother and Child Cohort

**DOI:** 10.3390/nu13030949

**Published:** 2021-03-16

**Authors:** Agnieszka Jankowska, Mariusz Grzesiak, Michał Krekora, Jolanta Dominowska, Joanna Jerzyńska, Paweł Kałużny, Ewelina Wesołowska, Irena Szadkowska-Stańczyk, Elżbieta Trafalska, Dorota Kaleta, Małgorzata Kowalska, Ewa Jabłońska, Beata Janasik, Jolanta Gromadzińska, Wojciech Hanke, Wojciech Wąsowicz, Gemma Calamandrei, Kinga Polańska

**Affiliations:** 1Department of Environmental and Occupational Health Hazards, Nofer Institute of Occupational Medicine, 91-348 Lodz, Poland; agnieszka.jankowska@imp.lodz.pl; 2Department of Perinatology, Obstetrics and Gynecology, “Polish Mother’s Memorial Hospital” Research Institute, 93-338 Lodz, Poland; mariusz.grzesiak@iczmp.edu.pl; 3Department of Gynecology and Obstetrics, IInd Chair of Gynecology and Obstetrics, Medical University of Lodz, 90-419 Lodz, Poland; michal.krekora@iczmp.edu.pl; 4Department of Obstetrics and Gynecology, “Polish Mother’s Memorial Hospital” Research Institute, 93-338 Lodz, Poland; 5Department of Teaching Midwifery, Medical University of Lodz, 90-419 Lodz, Poland; jolanta.dominowska@umed.lodz.pl; 6Department of Paediatrics and Allergy, Copernicus Memorial Hospital, Medical University of Lodz, 90-329 Lodz, Poland; joanna.jerzynska@umed.lodz.pl; 7Department of Environmental Epidemiology, Nofer Institute of Occupational Medicine, 91-348 Lodz, Poland; pawel.kaluzny@imp.lodz.pl (P.K.); ewelina.wesolowska@imp.lodz.pl (E.W.); irena.szadkowska-stanczyk@imp.lodz.pl (I.S.-S.); wojciech.hanke@imp.lodz.pl (W.H.); 8Department of Hygiene and Epidemiology, Medical University of Lodz, 90-752 Lodz, Poland; elzbieta.trafalska@umed.lodz.pl (E.T.); dorota.kaleta@umed.lodz.pl (D.K.); 9Department of Epidemiology, Medical University of Silesia, 40-752 Katowice, Poland; mkowalska@sum.edu.pl; 10Department of Translational Research, Nofer Institute of Occupational Medicine, 91-348 Lodz, Poland; ewa.jablonska@imp.lodz.pl; 11Department of Biological and Environmental Monitoring, Nofer Institute of Occupational Medicine, 91-348 Lodz, Poland; beata.janasik@imp.lodz.pl (B.J.); jolanta.gromadzinska@imp.lodz.pl (J.G.); wojciech.wasowicz@imp.lodz.pl (W.W.); 12Centre for Behavioural Sciences and Mental Health, National Institute of Health, I-00161 Rome, Italy; gemma.calamandrei@iss.it

**Keywords:** essential elements, vitamins, zinc, copper, selenium, diet, plasma, pregnancy, environmental determinants

## Abstract

The study objective was to identify determinants of essential elements and vitamins intake, and microelements and vitamins concentration in blood among pregnant women from Poland. Based on the data from food frequency questionnaires and information about supplements taken (*n* = 1252), daily supply of six elements (calcium, magnesium, iron, zinc, copper, selenium) and nine vitamins (folate, vitamins A, E, C, B1, B2, B3, B6, B12) was calculated. Zinc, copper, selenium (*n* = 340), vitamin A and E (*n* = 358) concentration was determined in blood collected during pregnancy. Most of the women did not meet the demand for essential elements and vitamins with a diet. About 94% of the respondents declared supplements use. The women with higher education, indicating leisure-time, physical activity and multiparity had a higher chance of meeting the average demand for the majority of the analyzed nutrients. On the other hand, factors such as BMI < 18.5kg/m^2^, a higher level of stress, and late first medical-care visit were associated with a lower chance of meeting the recommendations. Higher socio-economic status was a determinant of a higher selenium concentration in plasma (β = 3.1; 95%CI: 0.2–5.9), whereas BMI ≥ 25 kg/m^2^, and multiparity of a higher copper concentration in plasma (β = 0.2; 95%CI: 0.03-0.4; β = 0.2; 95%CI: 0.1–0.4). Higher plasma concentration of vitamin E was noted among women older than 30 years of age comparing to those who were 30 or younger (β = 1.5; 95%CI: 0.6–2.4). Although more studies are required, especially such based on laboratory measures, our results indicate target groups for dietary interventions during pregnancy for children’s optimal health and development.

## 1. Introduction

It is now well recognized that a diet is one of the most important lifestyle related factors that has influence on our health. The diet during pregnancy should be properly balanced and provide all the necessary nutrients. Both excess and deficiency of nutrients, as well as an inadequate supply of essential elements and vitamins, can result in diseases in mother and her offspring or decreased health and developmental potential of a child [1,2,3]. As an example, deficiencies of certain vitamins and minerals during pregnancy, such as vitamin E, C, B6, calcium (Ca), zinc (Zn), might play a role in pre-eclampsia, whereas iron (Fe) and Zn deficiencies are linked to impaired immunity [4]. It is also underlined that Fe, folate and vitamin A deficiencies can be associated with anaemia. Moreover, the role of adequate folates status before conception and at the beginning of pregnancy in the reduction of the risk of neural tube defects is scientifically proven [4,5]. 

The existing studies point out an association between maternal antenatal diet (defined as: (1) healthy vs. unhealthy based on the consumed food, (2) measured by dietary indices such as Dietary Approaches to Stop Hypertension (DASH) score and Dietary Inflammatory Index (DII) score or (3) evaluated based on individual nutrients intake or status) and birth outcomes, child adiposity, respiratory, cardiometabolic, and neurodevelopmental health [6,7,8,9,10,11,12,13]. This fully complies with the Developmental Origins of Health and Disease hypothesis, underlying that transient environmental exposures during critical periods of development (such as the pre-conceptional, fetal and early infant phases) can alter normal physiology and have a persistent impact on metabolism and gene expression, thereby influencing disease risk in later life and even intergenerationally [14].

While nutritional requirements of the general population can be achieved by an adequate diet, the risk of suboptimal nutrient intake (especially for Fe and folic acid) is common during pregnancy. Therefore, according to the guidelines developed by the World Health Organization (WHO) and The Polish Society of Gynecologists and Obstetricians (PSGO) some supplements might be needed to meet specific recommendations for the whole pregnancy or certain sensitive developmental stages [4,5,15,16]. Antenatal supplements that include folic acid and Fe are recommended by WHO in the context of rigorous research [4,5]. PSGO recommends the following folic acid supplementation: 0.4 mg/day before conception and 0.4–0.8 mg/day at the beginning of pregnancy and 0.6–0.8 mg/day after 12 weeks of pregnancy and during lactation. Moreover, assessment of blood count and ferritin concentration at the 1st prenatal visit and then blood count at 15–20, 27–32, 33–37 and 38–39 weeks of gestation is recommended. Fe supplementation before 16 weeks of pregnancy is advised among women with Fe deficiency anemia (Hb < 11 g/dL and low ferritin concentration) and after 16 weeks of pregnancy at a dose of up to 30 mg/day in women without anemia with the ferritin concentration below 60 μg/L [16]. 

Despite numerous studies confirming importance of a balanced diet during pregnancy, many pregnant women in the world, including those in the developed countries, are still at risk of suboptimal essential elements and vitamins intake [17]. Knowledge on factors that influence dietary choices and, as a consequence, nutritional status during pregnancy is crucial for the assessment of population needs and development of effective public health messages and interventions [18]. Most of the existing research indicates that better-educated women, those with higher socioeconomic status (SES) and more favorable lifestyle (physically active and nonsmokers) are more inclined to follow dietary recommendations and meet the need for essential nutrients [19]. However, some country-specific differences in the subpopulations’ adequate minerals and vitamins intake might exist. 

The aim of the study was to identify determinants of adequate antenatal intake of essential elements and vitamins from dietary and total (diet and supplements) sources, and determinants of selected microelements and vitamins concentrations in the blood collected during the 1st trimester of pregnancy. The mentioned data is crucial for recognition if pregnant women in Poland meet the demand for the essential nutrients and for identification of vulnerable groups for dietary interventions for children’s optimal health and development.

## 2. Material and Methods

### 2.1. Study Design and Population

The study is based on the data from the Polish Mother and Child cohort (REPRO_PL), which was established in 2007 (with the recruitment of women in the 1st trimester of pregnancy over 4 years). REPRO_PL is an ongoing population-based prospective cohort that has been created to investigate the effects of socio-demographic, lifestyle, environmental, and pregnancy-related variables on children’s health and development. Detailed information on the REPRO_PL cohort has been published elsewhere [20,21,22]. The cohort comprises 3 phases covering pregnancy, early childhood (12 and 24 months), and early school age (7 years) periods. The present analysis includes data that has been collected during pregnancy. 

In total, 1764 pregnant women were included into the REPRO_PL cohort [20]. Information regarding their diet and medications/supplements use was provided by 1508 women (86%) by completing questionnaires during the 1st and/or 2nd trimester of pregnancy. Of this group, 1306 women filled in the food frequency questionnaire (FFQ). Moreover, 202 women completed the 24 h Dietary Recall Questionnaire (24HR) and they were excluded from the current analysis. The final sample was restricted to 1252 women (96%) after exclusion of those with missing (*n* = 40) or inaccurate data (i.e., implausible energy intakes (<500 or >3500 kcal/day); *n* = 5) or women following special diets (i.e., vegetarian or vegan; *n* = 9) [23,24]. Assessment of microelements and vitamins concentration in blood plasma, which was collected during the 1st trimester of pregnancy, was performed on a subsample of the population (microelements: *n* = 340; vitamins: *n* = 358).

The study was approved by the Ethical Committee of the Nofer Institute of Occupational Medicine, Lodz, Poland (Decision No. 7/2007), and written consent was obtained from all the study subjects.

### 2.2. Assessment of Essential Elements and Vitamins Intake Based on Questionnaire Data

A detailed description of dietary data collection procedure and the FFQ has been published previously [25,26,27,28]. Briefly, the women, assisted by trained personnel, reported their consumption for a list of foods on a frequency scale with the following response categories: (a) never, (b) less than once per month, (c) 1–3 times per month, (d) 1–3 times per week (e) 4–6 times per week, and (f) every day. Based on the frequency of food products consumption and the size of an average portion, the content of 6 essential elements (calcium (Ca), magnesium (Mg), iron (Fe), zinc (Zn), copper (Cu) and selenium (Se)) and 9 vitamins (folate, vitamin A, E, C, thiamin (B1), riboflavin (B2), niacin (B3), pyridoxine (B6), cobalamin (B12)) in the consumed diet was calculated using the Polish Food Composition Tables [29]. These values were related to dietary standards, which allowed for the calculation of the percent of implementation of dietary recommendations (estimated average requirement (EAR) for the analyzed essential elements and vitamins is presented in Appendix A) [30]. 

During the 1st and 2nd prenatal visits, scheduled within the study, the women were interviewed by a gynecologist or midwife about the use of medications and supplements [20]. They provided the name, dose and period of their intake. Based on this information, the intake of essential elements and vitamins with supplements was calculated and included in the analyses. 

### 2.3. Determination of Microelements and Vitamins in Biological Samples

Blood samples taken from the women during the 1st trimester of pregnancy were centrifuged and plasma samples were collected, and then stored until the analyses. Concentrations of the selected microelements and vitamins were measured in the plasma samples using the methods that have been described in detail elsewhere [31,32]. Briefly, the flame atomic absorption spectrometry (FAAS) method was used for measurement of Zn and Cu [33] and the graphite furnace atomic absorption spectrometry (GFAAS) for Se concentrations in biological samples [34,35]. The use of the HPLC system integrated with the UV-VIS 190–800 nm detector allowed for the determination of vitamins A and E concentration in plasma [36].

### 2.4. Covariates

All the variables considered in the study were described in detail previously [27]. Briefly, the following socio-demographic covariates were assessed: maternal age, educational level, marital status, occupational activity between the 8th–12th weeks of pregnancy and socio-economic status (SES) (based on a subjective assessment as described by Polanska et al. 2017) [31]. In addition, two environmental factors, namely place of residence (based on the number of inhabitants) and seasonality (as the month of filling FFQ), were investigated. We also considered five variables related to lifestyle/health behaviour: maternal smoking status (with 10 ng/mL as a cut-off point for cotinine level in saliva being an indicator of active smoking as described by Wesolowska et al. 2019 and Stragierowicz et al. 2013) [27,37], alcohol consumption (yes and no), leisure-time physical activity (LTPA) (yes and no), the level of stress (based on the perceived stress scale (PSS) as described by Wesolowska et al. 2019) during pregnancy [27] and pre-pregnancy BMI (kg/m^2^). Moreover, the following pregnancy-related variables were assessed: parity, pregnancy-related adverse symptoms and complications and week of pregnancy of the 1st medical-care visit (based on medical records taking into account the date of the 1st medical-care visit and the date of the last menstrual period). Finally, the sex of a child was also considered. 

### 2.5. Statistical Analysis

Data regarding characteristics of the population, intake of essential elements and vitamins as well as microelements and vitamins concentrations in the blood samples are presented as numbers and percentages for categorical variables or means and standard deviations (SD) for continuous variables.

The level of consumption of the studied essential elements and vitamins were considered as a binary variable (coded as 0 = below the recommended level and 1 = equal or exceeding the recommended level). Intake from dietary sources and total intake from dietary sources and supplementation were considered as separate outcome variables. 

First, crude odds ratios (ORs) for attaining the recommended level of every essential element and vitamin in groups of mothers defined by each categorical determinant of interest was calculated by simple logistic regression models with one explanatory variable. Then, mutually adjusted ORs were calculated with a multiple logistic regression models. The outcome variable was considered for multiple logistic regression modelling if manageable number of subjects of each category were present in the study population (thus, folate and Fe from dietary sources were not presented for multivariable modeling). Each multivariable model comprised the following explanatory variables: maternal age (≤30 years (ref.), >30 years); maternal education (≤12 years of education (ref.), >12 years); occupational activity (no (ref.), yes); SES (low/middle (ref.), high); pre-pregnancy BMI (<18.5 kg/m^2^, 18.5 kg/m^2^–24.9kg/m^2^ (ref.), ≥25); LTPA (no (ref.), yes); PSS (<17 points (ref.), ≥17 points); place of residence (<100,000 inhabitants, ≥ 100,000 inhabitants (ref.)); a season of the year when the questionnaire was filled in (May–October, November–April (ref.)); parity (0 (ref.), ≥1 child); the 1st medical-care visit during pregnancy (≤6 weeks (ref.), > 6 weeks). Initially, more variables were considered as potential determinants: (marital status (married (ref.), unmarried); smoking (cotinine level in saliva <10 ng/mL (ref.), ≥10 ng/mL); alcohol consumption (no (ref.), yes); adverse symptoms and complications (no (ref.), yes); the sex of a child (male (ref.), female) but they were not included in the multiple logistic regression modelling. Either they consistently had no statistically significant crude effects, or confidence limits for multivariable ORs of these variables were not reliably estimable due to similarity to the outcome variables. The OR with 95% confidence intervals (95% CI), were presented as forest plots. Numerical values of OR and 95%CI from the univariable and multivariable models were also tabulated in Appendix A. 

To assess determinants of plasma concentrations of Zn, Cu, Se, vitamins A and E, these outcome variables were modeled using the ordinary linear regression with the same set of explanatory variables as the stated above. Regression coefficients β with 95% CI from the univariable and multivariable analyses were presented in the tables. Multivariable models for plasma concentrations omitted the following variables: marital status, adverse pregnancy symptoms and complications, and sex of a child, because none of the coefficients from the univariable analyses were below 0.1 significance level. 

The significance level of *p* = 0.05 was assumed in all the statistical procedures. The R software for statistical computing (version 3.5.1) was applied for statistical modeling and data processing; the R package forestplot (https://CRAN.R-project.org/package=forestplot; accessed on 20 December 2020) was used for plotting regression results [38]. 

## 3. Results 

### 3.1. Characteristics of the Study Population

Appendix A summarizes socio-demographic, lifestyle, environmental and pregnancy-related sample characteristics. The majority of women were 30 years of age or younger (64%), married (81%), declared a university degree (69%). One in four women assessed their SES as high (24%). Among the examined women, 55% were professionally active between the 8th and 12th weeks of pregnancy. About 18% of the respondents were classified as overweight or obese before their current pregnancy and 11% as active smokers during pregnancy. Moreover, 7% of the women indicated alcohol consumption and 69% reported LTPA during pregnancy. The 1st medical-care visit at the 6th week of pregnancy or earlier was reported by almost two-thirds of the women (59%) and about 4% of them noticed adverse pregnancy symptoms or complications up to the 2nd trimester of pregnancy. More than half of the women were in their first pregnancy (51%).

### 3.2. Factors Associated with Adequate Intake of Essential Elements and Vitamins during Pregnancy

Appendix A shows the percentage of pregnant women using dietary supplements and percentages of selected essential elements and vitamins from the supplements. A total of 94% of the respondents declared the use of supplements during pregnancy. The highest levels of supplementation were related to folate (100%) and vitamin B6 (50%). Essential elements supplemented to the lowest extent were Cu (7%) and vitamins B1 and B2 (2%).

Details on the intake of essential elements and vitamins from a diet and total consumption (from a diet and supplementation) are presented in Appendix A and Figure 1. The vast majority of the women (which in the current study represent more favourable socio-demographic status: university degree and high SES, marital status) did not meet the demand (referred to EAR) for most essential elements and vitamins along with a diet. As an example, inadequate intakes were noted for more than 80% of the study population for Mg, Fe and folate with the substantial improvement achieved with the supplementation of folate (93% of the population meet EAR from total consumption). Only for Cu, vitamins A, B2, B3, and B12, more than 70% of the women met the EAR with the consumed diet. 

Results of the univariable models for determinants of adequate intake of selected essential elements and vitamins from dietary and total (dietary and supplements) sources are presented in Appendix A. The data from relevant multivariable models are presented in Appendix A as well as in a form of a forest plot (Figure 2). The women with a university degree had a higher chance of achieving adequate intake of essential elements and vitamins from dietary sources (Ca: OR = 1.5; 95%CI: 1.1–2.1; Mg: OR = 1.6; 95%CI: 1.1–2.3; Zn: OR = 1.4; 95%CI: 1.1–1.9; Se: OR = 1.5; 95%CI: 1.1–2.0; vitamin B1: OR = 1.4; 95%CI: 1.0–1.8; vitamin B3: OR = 1.4; 95%CI: 1.0–1.8)) comparing to those with a lower educational level (Figure 2; Appendix A). Moreover, a higher educational level was a determinant of an adequate status for folate (OR = 2.8; 95%CI: 1.7–4.6) (Figure 2 and Appendix A). The older women had a higher (OR = 1.4; 95%CI: 1.1–1.9) and those working during pregnancy a lower (OR = 0.7; 95%CI: 0.5–0.9) chance of adequate Se intake with dietary sources comparing to the younger and unemployed ones (Figure 2 and Appendix A). The women declaring high SES achieved EAR with dietary sources for vitamin C (OR = 1.5; 95%CI: 1.2–2.0) more frequently as compared to the women with lower SES (Figure 2 and Appendix A). The women classified as underweight before pregnancy had a lower chance of achieving vitamin B2 (OR = 0.5; 95%CI: 0.3–0.9) and B12 (OR = 0.4; 95%CI: 0.2–0.7) requirements with a diet, and Zn (OR = 0.6; 95%CI: 0.4–1.0) with total (dietary and supplements) sources (Figure 2, Appendix A) compared to those within the recommended BMI category. Recreational physical activity during pregnancy was a strong determinant of adequate essential elements and vitamins intake (*p* < 0.05) (Figure 2 and Appendix A). On the other hand, a higher number of points from PSS (indicating a higher level of stress) was associated with a lower chance of achieving EAR for some essential elements (Cu, Se) and vitamins (C, B3, B6, B12) (*p* ≤ 0.05)) (Figure 2 and Appendix A). The place of residence was not a significant determinant of an adequate essential elements and vitamin status. The multiparous mothers had a higher chance to meet EAR recommendations for Cu (OR = 1.5; 95%CI: 1.1–2.1), Se (OR = 1.5; 95%CI: 1.2–2.0), vitamin A (OR = 1.7; 95%CI: 1.1–2.6) and vitamin B6 (OR = 1.4; 95%CI: 1.0–2.0—for total: dietary and supplement sources) as compared to the nulliparous women (Figure 2 and Appendix A). Finally, a late 1st medical-care visit was associated with a lower chance of achieving EAR from total (dietary and supplements) sources for Se (OR = 0.7; 95%CI: 0.6–1.0), folates (OR = 0.5; 95%CI: 0.3–0.8) and vitamin A (OR = 0.7; 95%CI: 0.4–1.0) (Figure 2 and Appendix A). 

### 3.3. Factors Associated with Microelements and Vitamins Concentration in Blood Collected during Pregnancy

The mean (±SD) microelements and vitamins concentration in blood (plasma) collected during the 1st trimester of pregnancy is presented in Appendix A. Appendix A (the univariable linear regression model) and Table 1 (the multivariable linear regression model) show determinants of micronutrients and vitamins concentration in plasma. Based on the multivariable linear regression model, higher SES was a significant determinant of a higher Se concentration (β = 3.1; 95%CI: 0.2–5.9) and BMI ≥ 25 kg/m^2^ as well as multiparity of a higher Cu concentration (β = 0.2; 95%CI: 0.03–0.4 and β = 0.2; 95%CI: 0.1–0.4, respectively). Higher concentration of vitamin E was noted among the women older than 30 compared to those who were 30 or younger (β = 1.5; 95%CI: 0.6–2.4).

## 4. Discussion

The presented study indicates that the vast majority of women in Poland did not meet the demand for the most essential elements and vitamins with the food they consumed. Moreover, despite widespread use of supplements in the analyzed population, still in the case of some crucial minerals and vitamins the recommendations were not met. The assessments based on the questionnaire data show that a higher chance of meeting the average demand for majority of the analyzed nutrients was observed among the women with a higher educational level, declaring LTPA during pregnancy and multiparity, whereas factors such as: being underweight, experiencing a higher level of stress or late the 1st medical-care visit were associated with a lower chance of meeting the recommendations. Moreover, the analysis based on microelements and vitamins concentration in blood collected during the 1st trimester of pregnancy indicates a higher SES as a determinant of a higher Se concentration, BMI ≥ 25 kg/m^2^ as well as multiparity as a determinant of a higher Cu concentration and older age (>30 years) as a determinant of a higher vitamin E concentration. We believe that our results have important implications for health professionals and public health authorities as well as for future research. They might help support adequate dietary practices before and during pregnancy for children’s optimal health and development. 

According to the current recommendations, pregnant women should consume a healthy, balanced diet consistent with guidelines on healthy eating to guarantee the right amount of energy and nutrients, as well as an adequate supply of vitamins and minerals [4,5,39]. What is important is that, in line with the statements of WHO and majority of medical associations, a routine usage of dietary supplements by all pregnant women is not recommended [4,5,15,16]. Moreover, it needs to be underlined that some disparities exist in the recommended level of nutrients intake across the countries, which can result from scientific evidence available when the recommendations were developed or criteria selected for the development of specific guidelines [17]. 

In our study, about 94% of the pregnant women declared supplements use. The estimates from the developed countries, including USA and countries in Europe, indicate that prenatal dietary supplementation has been used by 70–98% of the women [40,41,42]. Similarly to the data from other studies, our analysis confirms that folic acid is the most commonly taken supplement. The EuroPrevall cohort study conducted in nine European countries shows the uptake of folic acid in Poland by 80% of women (the highest uptake was observed in Spain and the UK: 98% and 88%, respectively, and the lowest in Lithuania: 56%) [40]. Fe supplementation is recommended in Poland under specific circumstances (as it was described previously) and our data indicate that it is taken by 24% of the pregnant women. These percentages are lower compared to other data from Poland, where 35% women declared Fe supplementation [43]. It is worth mentioning that the slight differences between studies may be partially a result of different gestational ages within study populations. 

As it was underlined previously, a healthy and balanced diet should guarantee all necessary nutrients to a developing fetus. However, our assessments and the data from other studies (including those conducted in Poland) indicate that for some recommended values for minerals and vitamins, with important roles during pregnancy, nutritional adequacy has not been achieved [44,45,46]. This indicates that more attention should be paid to a better knowledge on the principles of a healthier diet in accordance with the existing recommendations. Polish guidelines, which are in line with that developed by WHO, advise eating meals regularly and composing them using products from all groups (vegetables and fruit; wholegrains; dairy; meat; fish; eggs; vegetable oils and legumes). Salty snacks, dairy sweets drinks and deserts that have typically been characterized as these with poor nutritional value, containing mainly fat and sugars should be limited. Moreover, processed food should be avoided [39,47,48]. Our analysis, similarly to other studies in this field, shows that supplements can increase but not always ensure the recommended minerals and vitamins intake [41,44]. It needs to be underlined that some nutrients were received cumulatively and in large amounts from different supplements, whereas at the same time some relevant supplementation was missing. Bioavailability of microelements in different supplementation products was not considered, and remains a possible source of uncertainty in our estimates of total intakes including supplementation. Nevertheless, considering all existing data, dieticians, nutritionists, physicians and other healthcare providers should be able to offer accurate and evidence-based advice on supplement use in pregnancy. Routine supplementation may not be necessary for all women, and in some cases it may even carry a health risk, thus, laboratory data could be required for appropriate supplementation planning and management [42,49].

There are several factors, both environmental and lifestyle-related, that can determine dietary choices, supplement use as well as essential elements and vitamin levels in the body [44,50,51,52,53,54,55,56,57,58,59,60]. A lower educational level and SES of the family may be a risk factor for an unhealthy lifestyle, including an improper diet, obesity or smoking [59,61]. Moreover, women with a lower educational level and SES usually schedule their 1st medical-care visit later in pregnancy and meet a gynecologist or midwife less often, which decreases their chances for dietary assessments and modifications as well as adequate supplement use [7]. This has been also observed in our current study (and similarly in a previously published analysis that was based on dietary patterns) where the women with higher education and indicating recreational physical activity represented better dietary choices and, as a consequence, had a higher chance of meeting the average demand for the majority of the analyzed nutrients [27]. On the other hand, the late 1st medical-care visit was associated with a lower chance of meeting the recommendations. It needs to be underlined that in this study, we have categorized the 1st prenatal visit as before or equal 6 weeks of gestation and after 6 weeks of gestation. Setting the cut-off point for an early prenatal check-up at week 6 of gestation seems to be adequate in terms of doing it as early as possible, but it may be biased as some of the women (especially these with irregular menstrual period) in that week may be unaware of their state of pregnancy. Taking this into account, the week of pregnancy of the 1st medical-care visit has been additionally considered as a continuous variable in the analyses (data not shown). The results were similar to those obtained in the previous models — later the 1st medical-care visit translated into a lower chance of meeting the average demand for majority of the analyzed nutrients.

What is interesting is that in the present assessments the multiparous women had a higher chance of meeting the average demand for some of the nutrients. To date, findings on how parity can impact nutritional choices are not fully conclusive [19,62,63,64,65]. It is possible that parity can be a marker for socio-demographic factors or that the influence of parity on dietary choices may depend on a wider context (e.g., support allocated to the women in their first pregnancy and those who already have children) [19]. Moreover, mothers who already have children may pay more attention to the quality of meals prepared for the whole family and, therefore, they may choose food with an increased nutritional value. 

A detailed discussion of the micronutrient and vitamin concentrations in the biological samples observed in our population has been published elsewhere [31,32,66]. It needs to be underlined that micronutrient and vitamins concentrations can vary depending on the pregnancy period. Moreover, they can also vary worldwide from low to even toxic concentrations [67,68,69,70,71]. In our population, higher SES was a determinant of a higher Se concentration in blood, which is in agreement with the questionnaire data where a higher educational level and SES were related to higher Se intake with a diet. Similarly, a higher Cu concentration in the blood and higher intake with a diet and total (diet and supplements) was observed among the multiparous women. We also observed a higher Cu concentration among overweight women compared to those within the recommended weight category. This observation is consistent with other studies showing a generally positive association between serum Cu level and obesity (a recent meta-analysis by Gu et al., 2020), or a positive association between Cu level in serum or adipose tissue and BMI [72,73], which, altogether, suggest that Cu status is strongly related to BMI, independently of pregnancy. Interestingly, Yang et al. (2019) have found that total Cu levels positively correlate with leptin, insulin, and leptin/BMI suggesting that Cu and/or cuproproteins may be functionally linked to fat accumulation [73]. Nevertheless, the role of Cu in obesity has not been elucidated yet and, thus, it is not possible to conclude about adequacy of Cu status in overweight pregnant women and make any special dietary recommendations concerning Cu intake in this group. The vitamin E mean concentration noted in our population (8.2 mg/L ± 3.6 mg/L) is similar to that reported by Chen et al. (2018) (9.1 mg/L ± 2.5 mg/L) [74]. Higher vitamin concentration among older women seems to be rather related to their dietary choices not to supplementation.

The prospective study design with a reliable assessment of several lifestyle or pregnancy-related factors (i.e., a cotinine level in saliva as a biomarker of smoking status, pregnancy complications, or supplement use evaluated by a gynecologist or midwife during a detailed interview) is an important strength of the current study. Additionally, our assessments were based on a medium-to-large sample size population. It also needs to be pointed out that the study was restricted to the healthy women, in healthy pregnancy not assisted with reproduction technologies, which, in principle, means that the women did not require special diets or medical treatment. Moreover, we evaluated a set of essential elements and vitamins intake based on the questionnaire data and also for some minerals and vitamins based on their concentration measured in blood. Consumption of supplements was also taken into account. Finally, we looked at a variety of determinants (socio-demographic, lifestyle-related, environmental, and pregnancy-related factors) of an adequate essential elements and vitamin status during pregnancy, which can provide more accurate information for the development of dietary interventions targeting vulnerable groups. Limitations of the study are related to the assessment of dependent variables, which was mostly based on FFQ data (for six minerals and nine vitamins) and only for a few nutrients also based on laboratory assessments (for three minerals and two vitamins). Thus, over or underreporting of nutrients intake could occur. Moreover, declared food intake may also not correspond to the nutritional status measured by laboratory means as a result of different bioavailability of nutrients from different food products and individual differences in metabolism. Even though questionnaire data is typical of public health research, both measures are needed for appropriate assessment of a nutritional status and to support the conclusions drawn within this study [51,65,75]. To minimize the measurement error when assessing dietary intake with FFQ, future studies should be eventually validated using duplicate portions technique [76]. The study by Thijsburg et al. indicated that duplicate portions technique was less influenced by proportional scaling bias, and correlated errors between duplicate portions technique and FFQ were lowest compared with the 24HR. Attenuation factors also indicated that the duplicate portions technique performed better than the 24HR. Therefore, the authors conclude that the duplicate portions technique is probably the best available reference method for FFQ validation for nutrients that currently have no generally accepted biomarker. We did not collect data at several time points, simultaneously making it impossible to identify any changes in dietary patterns throughout the pregnancy period. Moreover, even though we have evaluated a wide range of potential determinants of nutritional status, some food policies and recommendations or weight concern, which all can impact the women’s decisions regarding dietary choices were omitted. Finally, the data presented in this manuscript are from the phase of the study that was performed between 2007 and 2011 so the pattern of a diet and supplement use might have changed over the time. In this respect, more updated data are needed-ideally presenting the results from validated questionnaires as well as biological sample assessments.

## 5. Conclusions

Diet of pregnant women, despite existing recommendations, is still significantly lacking balance for specific minerals and vitamins crucial for the proper development of a fetus and children’s health. Further efforts to encourage women to establish healthy dietary practices before conception and throughout the pregnancy period are warranted. The less educated women, and those with coexisting less favorable lifestyles should be target groups for such actions. Promotion of the 1st medicalcare visit as early during pregnancy as possible also creates a possibility to monitor nutritional status of the women early and introduce promotional and educational interventions. Finally, more studies are required, especially based on laboratory measures, to provide a more reliable assessment of nutritional status during pregnancy and to implement adequate preventive measures. The duplicate portions technique can be considered as the best available reference method for FFQ validation in the case that laboratory measurements are not possible.

## Figures and Tables

**Figure 1 nutrients-13-00949-f001:**
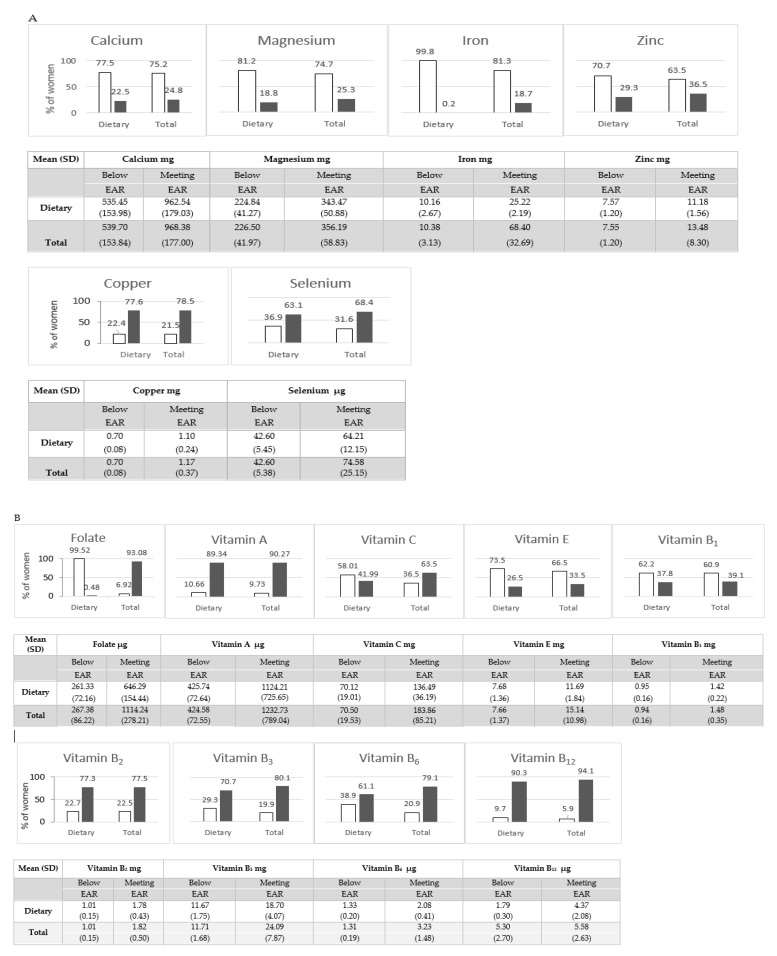
Intake of essential elements (**A**) and vitamins (**B**) from diet and total intake (from diet and supplementation) according to daily intake category: 
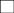
 Below EAR, 
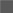
 Meeting EAR. Note: EAR (Estimated Average Requirement) for essential elements and vitamins during pregnancy are presented in Appendix A.

**Figure 2 nutrients-13-00949-f002:**
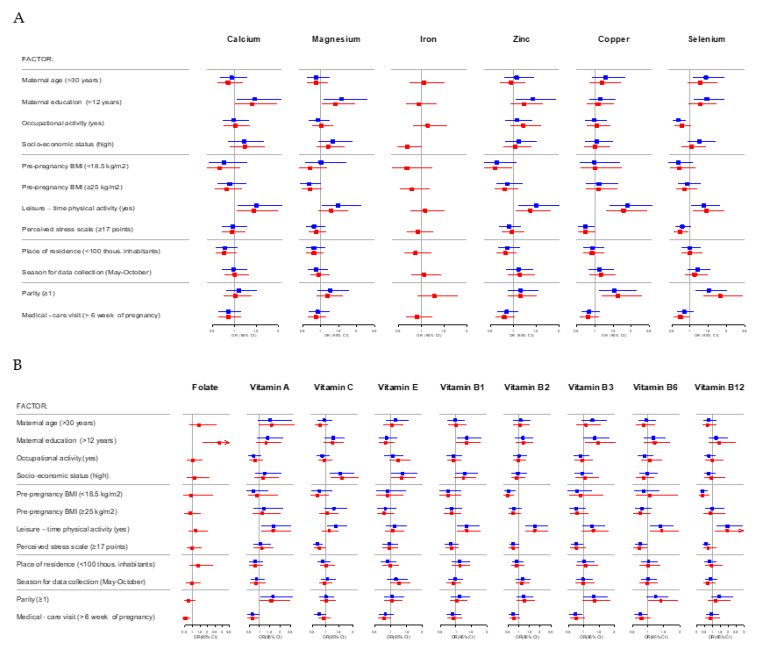
Adjusted effects of the explanatory factors for the adequacy of intake of selected essential elements (**A**) and vitamins (**B**) from dietary (blue) and total (dietary and supplementation) (red) sources, estimated by multivariable logistic regression models. For iron intake from dietary sources multivariable modelling was not feasible, because only 2 mothers attained recommended level. For folate intake from dietary sources multivariable modelling was not feasible, because only 6 mothers attained the recommended level. Reference groups: maternal age ≤ 30 years; maternal education ≤ 12 years; occupational activity between the 8th–12th weeks of pregnancy—no; SES—low/medium; pre-pregnancy BMI (kg/m^2^)—18.5–24.99; LTPA—no; PSS < 17 points; place of residence (thousands of inhabitants) ≥ 100; season for data collection—November–April; parity—0; week of pregnancy of the 1st medical-care visit ≤ 6.

**Table 1 nutrients-13-00949-t001:** Adjusted effects of the explanatory factors on microelements and vitamins concentrations in blood plasma samples collected during the 1st trimester of pregnancy estimated by multivariable linear regression models.

Determinant	Zinc (mg/L)	Copper (mg/L)	Selenium (µg/L)	Vitamin A (mg/L)	Vitamin E (mg/L)
β (95% CI)					
**Maternal age**					
>30	0.04 (−0.03, 0.10)	−0.08 (−0.22, 0.07)	2.56 (−0.15, 5.27)	0.01 (−0.06, 0.08)	1.51 (0.62, 2.39) *
**Maternal education (years)**					
>12	−0.05 (−0.12, 0.02)	0.04 (−0.11, 0.19)	−0.98 (−3.76, 1.80)	0.01 (−0.06, 0.09)	−0.23 (−1.14, 0.68)
**Occupational activity between the 8th and 12th week of pregnancy**					
Yes	−0.06 (−0.12, 0.01)	−0.09 (−0.23, 0.04)	0.22 (−2.30, 2.73)	0.09 (−0.03, 0.16)	−0.32 (−1.12, 0.47)
**Socio-economic status (SES)**					
High	0.03 (−0.04, 0.11)	−0.04 (−0.19, 0.12)	3.07 (0.21, 5.94) ^^^	0.03 (−0.04, 0.09)	−0.35 (−1.22, 0.52)
**Pre-pregnancy BMI (kg/m^2^)**					
<18.5	0.01 (−0.10, 0.12)	-0.11 (−0.35, 0.14)	−0.89 (−5.46, 3.68)	−0.08 (−0.21, 0.05)	−1.41 (−3.00, 0.18)
≥25	0.07 (−0.00, 0.15)	0.19 (0.03, 0.35) ^^^	1.98 (−1.04, 5.00)	0.05 (−0.03, 0.13)	−0.28 (−1.25, 0.70)
**Cotinine level**					
≥10 ng/mL	0.05 (−0.05, 0.15)	0.05 (−0.17, 0.27)	−2.29 (−6.35, 1.77)	0.03 (−0.08, 0.13)	0.49 (−0.82, 1.80)
**Alcohol consumption**					
Yes	0.01 (−0.09, 0.11)	−0.02 (−0.24, 0.20)	−2.30 (−6.33, 1.73)	0.08 (−0.03, 0.19)	1.28 (−0.10, 2.66)
**Leisure–time physical activity (LTPA)**					
Yes	0.05 (−0.01, 0.12)	0.04 (−0.10, 0.18)	2.45 (−0.14, 5.04)	0.03 (−0.03, 0.10)	−0.24 (−1.06, 0.59)
**Perceived Stress Scale (range: 0–38 points)(PSS)**					
≥17 points	0.07 (0.00, 0.13) ^^^	−0.05 (−0.18, 0.08)	−0.40 (−2.85, 2.05)	−0.03 (−0.09, 0.03)	0.22 (−0.56, 0.99)
**Place of residence (thousands of inhabitants)**					
<100	−0.06 (−0.13, 0.01)	0.09 (−0.06, 0.24)	0.29 (−2.51, 3.09)	−0.09 (−0.16, 0.02)	0.21 (−0.63, 1.06)
**Season for data collection**					
May–October	0.01 (−0.06, 0.07)	0.04 (−0.09, 0.17)	2.03 (−0.39, 4.46)	−0.04 (−0.10, 0.02)	0.70 (−0.06, 1.46)
**Parity**					
≥1	0.01 (−0.05, 0.08)	0.21 (0.07, 0.36) *	0.18 (−2.46, 2.82)	−0.00 (−0.07, 0.06)	−0.61 (−1.45, 0.24)
**Week of pregnancy of the 1st medical-care visit**					
>6	0.02 (−0.05, 0.08)	−0.01 (−0.15, 0.12)	0.58 (−1.92, 3.09)	0.02 (−0.05, 0.08)	−0.38 (−1.18, 0.41)

Reference groups: maternal age ≤ 30 years; maternal education ≤ 12 years; occupational activity between the 8th–12th week of pregnancy—no; SES—low/medium; pre-pregnancy BMI (kg/m^2^)—18.5–24.99; cotinine level < 10 ng/mL; alcohol consumption-no; LTPA-no; PSS < 17 points; place of residence (thousands of inhabitants) ≥ 100; season-November–April; parity-0; week of pregnancy of the 1st medical-care visit ≤ 6. ^ *p* < 0.05; * *p* < 0.01, Each coefficient β represents expected increase/decrease in microelement or vitamin concentration, associated with the determinant, relative to the reference group, adjusted for all other determinants listed in the table.

## Data Availability

The data presented in this study are available on request from the corresponding author. The data are not publicly available due to privacy and ethical restrictions.

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
