# Peer review of "Determinants of the Essential Elements and Vitamins Intake and Status during Pregnancy: A Descriptive Study in Polish Mother and Child Cohort"

_nutrients, 2021, doi:10.3390/nu13030949_

Round 1

Reviewer 1 Report

The article “Determinants of an Adequate Essential Elements and Vitamins Status during Pregnancy “submitted by Agnieszka Jankowska estimates nutritional supply of pregnant women, largely on the use of FFQ and Dietary Recalls.

The topic is of interest, as pregnant women have special dietary requirements that need to be met to prevent damage to the growing fetus as well as to prevent the development of certain diseases later in live of the offspring.

Due to weaknesses in language and style, some parts of the explanations are unfortunately hard to understand. Sentences that need attention are marked in the manuscript, but the complete text needs to be revised

In addition, I have some major concerns:

  • A strong weakness of the article is that only for zinc, copper and selenium actual laboratory data are included. For all other elements, the intake was calculated from questionnaires. This is only an estimate and subjected to the error of answering the questionnaire / the compliance of the participants. Furthermore, as stated by the authors themselves, “The bioavailability of microelements in different supplementation products was not considered and remains [a] possible source of uncertainty in our estimates of total intakes including supplementation.” It is well known that requirements can vary by far between individuals and might also depend on the physical activity. Thus, adding laboratory data is mandatory to support the conclusions drawn here. As a similar approach was already undertaken for zinc and iron (Reference 34), the only rather new data might be those for selenium.
  • Background information provided in the introduction is often unspecific / too general. Details can be found in the comments to the manuscript.
  • Also, not all the recommendations that are described in the article are supported by information provided via the WHO. Please re-evaluate the literature. E.g. Vitamin D: How is the recommendation for Poland? In countries with a dark season such as Finland, supplementation is only recommended for pregnant women from October to April (as can be found in reference 35, cited in this study). It is difficult to extrapolate a low Vitamin D intake from food and supplements to a deficiency in Vitamin D. As no analyses of any symptoms of Vitamin D deficiency were included into this study, data for Vitamin D should be discussed separately from the others and carefully. The current conclusion is not supported by the data shown here. This statement is supported by the WHO recommendation, which has also been cited in the article (Reference 4, see the “Nutritional interventions update: vitamin D supplements during pregnancy “: “WHO recommendation on antenatal vitamin D supplements Oral vitamin D supplementation is not recommended for all pregnant women to improve maternal and perinatal outcomes. (Not recommended) Remarks • This recommendation updates and does not alter the WHO recommendation found in the WHO ANC guideline (1). • Pregnant women should be encouraged to receive adequate nutrition – which is best achieved through consumption of a healthy, balanced diet – and to refer to guidelines on healthy eating (2). • Pregnant women should be advised that sunlight is the most important source of vitamin D. The amount of time needed in the sun is not known and depends on many variables, such as the amount of skin exposed; the time of day, latitude and season; skin pigmentation (darker skin pigments synthesize less vitamin D than lighter pigments); and sunscreen use (3). • For pregnant women with suspected vitamin D deficiency, vitamin D supplements may be given at the current recommended nutrient intake (RNI) of 200 IU (5 µg) per day (1,4). This may include women in populations where sun exposure is limited.”
  • As mentioned in the article on a similar study in Finland, cited by the authors, some of the elements only require supplementation, in certain situations (e.g. if “If the Hb value is lower than 110 g/l during the first trimester of pregnancy, 50 mg supplemented Fe is recommended.). Are those recommendations similar in Poland? If yes, this should be included in the article and also when drawing a conclusion on the sufficient supply of the nutrients to the women. As the authors state themselves, over-supplementation can cause side effects (see for example https://www.annualreviews.org/doi/abs/10.1146/annurev-nutr-082018-124213) and those need to be discussed in the article as well. Here, again the article by X et al., is rather unspecific and only states that “Both excess and deficiency of nutrients, as well as an inadequate sup-ply of essential elements and vitamins, can result in diseases or decreased health and developmental potential of a child”. This should not be generalized as it is right now, or better defined for each element.
  • The authors found that subjects with a higher BMI have higher basal copper values as found in serum. What does this mean in regard to the question that this article addresses? Individuals with a high BMI do not need copper supplementation? Do pregnant women with a high BMI need their own nutritional recommendation? This would be in line with the suggested need of laboratory data for all elements, as subjects with a high BMI might not only reveal differences regarding the copper status compared to more lean pregnant women.
  • The following results/conclusion (although not exactly the same data) were already published in 2018 (Ref 34) and are thus not new: “Low levels of Fe, Zn, Ca, Mg, vitamin D, and folic acid intake were seen in the pregnant women, with the use of dietary supplements significantly increasing their intake of Fe, Zn, and folic acid.”. Please delete any overlap with the already published data and concentrate on only the novel data!
  • The authors state that “It also needs to be pointed out that the study is restricted to healthy women (excluding vegetarian or vegan women)…”. This implies that vegetarians or vegans are not healthy, which is not proven! A vegetarian diet might even be healthier than a non-balanced diet based on meet from fast food. Please revise!
  • Conclusion: Although it might be helpful to assess the nutritional status of pregnant women more closely, if their BMI is <18.5kg/m2, they have a higher level of stress, or were late with a first medical-care visits, laboratory data are necessary to proof this hypothesis. Laboratory data would probably be needed to plan the supplementation scheme anyways. Thus, the conclusion is not clearly supported by the data.
  • The discussion is weak. It in large parts again described the results. If other studies are mentioned, details on study design and outcome are missing and thus a comparison with the data from the authors is difficult.
  • The results are overinterpreted as laboratory data are missing. The conclusions should be formulated much more carefully.

Minor:

  • There are some redundant data / similar information can be found in introduction, results and discussion. Please delete any redundancy.

Author Response

Dear Reviewer,

Thank you very much for giving us a chance to improve our manuscript entitled “Determinants of an Adequate Essential Elements and Vitamins Status during Pregnancy” by Agnieszka Jankowska et al.

We highly appreciate all the valuable comments and we have made relevant amendments in accordance with your recommendations. We feel convinced that as a result of these modifications our manuscript has been considerably improved. Now, we are resubmitting the revised paper together with a point-by-point response below.

Best regards,

Kinga Polanska

Reviewer 2 Report

Thank you for this well-written manuscript assessing nutrient intakes during pregnancy.

While this is an interesting study, how does it relate to the diet of these pregnant women? It would be more infinitely meaningful to characterize the diets that are supplying adequate/inadequate nutrient intakes, even if it is with broad strokes i.e. typical Polish diet, Westernized, plant-based, etc.?

It would be useful to include some numbers in the results and/or conclusion with the estimated portion of this population that did not meet the EAR with and without the use of supplements. For example, it is notable in this sample (a sample in which the women seem to have family support/married and are educated) that inadequate intakes are near ~50% for vitamin C, ~60% for vitamin D and ~70-75% for calcium and magnesium.

Supplement use, particularly for vitamin D which is only based on supplemental intake, should be characterized.

Some minor grammatical issues: E.g. p. 9 footnote should read "2 mothers and these varialbes were not included" and p.12 3rd paragraph "folate supplementation"

Author Response

(The authors gave the same response as above.)

Reviewer 3 Report

Manuscript ID: nutrients-1068354

Title: Determinants of an adequate essential elements and vitamins status: a descriptive study in Polish mother and child cohort.

The aim of the study was to identify determinants of adequate antenatal intake of essential elements and vitamins from dietary and total (diet and supplements) sources, and determinants of selected microelements and vitamins concentrations in the blood collected during the 1st trimester of pregnancy.

Comments and Suggestions for Authors:

The manuscript is a very interesting study, but requires some considerations.

Page 3. "The cohort comprises 3 phases covering pregnancy, early childhood (12 and 24 months), and early school age (7 years) periods. The present analysis includes data that has been collected during pregnancy". However, the study is based on the data from the Polish Mother and Child cohort (REPRO_PL), which was established in 2007 (with the recruitment of women in the 1st trimester of pregnancy over 4 years). Why have you taken so long to analyze the data that was collected during pregnancy? Please clarify.

Page 3. “Moreover, 202 women completed the 24-hour Dietary Recall Questionnaire (24HR) and they were excluded from the current analysis”.  Why were they excluded?

Page 4. “During the 1st and 2nd perinatal visits, scheduled within the study, the women were interviewed by a gynecologist or midwife about the use of medications and supplements”. It should say “prenatal visits”.

Page 5. “The level of consumption of the studied essential elements and vitamins were considered as a binary variable (coded as 0 = below the recommended level and 1 = equal or exceeding the recommended level)”. Why has this variable not been treated in a staggered manner: by tertiles, quartiles or quintiles?

Page 5. “The first medical care visit during pregnancy (≤6 weeks (ref.), > 6 weeks)”. Setting the cut-off point for an early prenatal check-up at week 6 of gestation is adequate in terms of doing it as early as possible, but it may be a bias. It must be taken into account that pregnant women in week 6 of amenorrhea (as it must be considered) have been pregnant for only 4 weeks and with knowledge due to lack of menstrual period only 2 weeks. Likewise, you should take into account women who have long menstrual cycles (> 28 days) who in that week may be unaware of their state of pregnancy. All of this should be taken into account. First of all, you should clarify if it has been considered that the weeks of gestation are considered as amenorrhea (as is usually done). Secondly, the advisability of considering early gestational control at 6 weeks' gestation should be discussed.

Page 7. “However, those results need to be interpreted with caution as it needs to be taken into account that vitamin D is mainly produced by human body from exposure to sunlight”. This paragraph should not be in the Results section. It should be moved and discussed in the Discussion section.

Page 7. “Based on the multivariable linear regression model, higher SES was a significant determinant of a higher Se concentration (β=3.1; 95%CI: 0.2-5.9) and BMI ≥25kg/m2... Higher concentration of vitamin E was noted among the women older than 30 compared to those who were 30 or younger (β=1.5; 95%CI: 0.6-2.4)”. These results are given as statistically significant, but since the confidence intervals include the value 1, they should be checked.

Page 16. “To minimize the measurement error when assessing dietary intake with FFQ, future studies should be eventually validated using duplicate portions technique”. This clarification from the authors is very important and should be emphasized.

Author Response

Dear Reviewer,

Thank you for all your work on our manuscript entitled “Determinants of the Essential Elements and Vitamins Intake and Status during Pregnancy: a Descriptive Study in Polish Mother and Child Cohort” by Agnieszka Jankowska et al..

Your comments and suggestions were very useful and helped us to improve the paper considerably. All your suggestions have been taken into account in the recent revision of the manuscript.

Best regards,

Kinga Polanska

Below you can find responses to each of your specific comments.

Round 2

Reviewer 1 Report

The authors “nutrients-1068354 “ provide answers to the points mentioned during the previous review. Especially the language was improved. However, not all points were addressed sufficiently, and the article still lacks laboratory data and novel findings. For those existing laboratory data, significance is low especially in regard to the data derived from the questionnaires. Without those data, the novelty of the results and conclusions is very low.

The authors have added plasma data for Vitamin E and Vitamin A. However, the only significant correlation that was found was for pregnant women older than 30 years of age regarding Vitamin E. As with most of the significant correlations that were found when analyzing the laboratory data, the significant effects here are not mirrored by the findings derived from the intake data presented in figures 2 and 3.

That the quality of nutrition can be correlated to education level etc. has been suggested before (https://pubmed.ncbi.nlm.nih.gov/30789806/; https://pubmed.ncbi.nlm.nih.gov/31920469/; https://pubmed.ncbi.nlm.nih.gov/31847928/)  . It is also not new, that subjects with BMI higher or lower than the healthy range have a lower quality of nutrition. The parameters that were assessed in addition to previous studies and the higher number of women tested to not present a significant novelty and improvement compared to existing studies.

The results are not helpful for physicians regarding the consultation of pregnant women. Physicians should suggest adequate nutrition to all pregnant women, independent from for example educational or socio-economic status, which has been found before (https://pubmed.ncbi.nlm.nih.gov/31920469/). Moreover, especially in regard to iron, supplementation is only advisable, if a deficit is found when status is assessed via laboratory analyses.

Finally, there are no data, if the findings reported in the article are associated with any parameters related to pregnancy outcomes and children´s health. This is another major weakness of the study. If no effects of the decreased nutritional intake, which the authors suggest for some subgroups of their study population, appear in reality, there is no benefit of the data.

The authors comment: “We want to underline that REPRO_PL cohort was designed to study associations between environmental and lifestyle factors (phthalates, BPA, PAH, heavy metals, smoking, physical activity, diet, etc.) on pregnancy outcomes and children’s health. Thus, this study was not strictly designed to look at different micronutrient and vitamin levels in biological samples.”

If it was not the purpose, the title of the manuscript: “Determinants of an Adequate Essential Elements and Vitamins Status during Pregnancy” is strongly misleading Addressing one of my further comments, the authors changed the title: “Considering your valuable comment, we have revised the title of the manuscript, in order to make it more consistent with the aim of the study and the scope of analyses (we have changed the title into “Determinants of Essential Elements and Vitamins Intake and Status during Pregnancy” to emphasize that the study was focused on both dietary intake and status)”, which is still misleading. Only laboratory data can determine if an adequate intake has been achieved.

The authors argument that the parameter of for example place of residence and seasonality is a strength of the manuscript compared to similar studies. However, those points are not adequately evaluated and discussed. For example, it can be expected that the supplementation with Vitamin D is lower during summer months; probably physicians do even not recommend the supplementation during the period, unless symptoms of vitamin D deficiency appear. Sufficient Vitamin D is usually provided by sunlight during this time. The authors mention all this in the new part that was added to the manuscript, but do not discuss this new information sufficiently with their data. The authors mention that Vitamin D status is difficult to estimate themselves; thus, I do not see the benefit of trying it with a questionnaire as in the current article. The part on vitamin D does thus not provide any useful data and should be deleted.

The author’s data even suggest Vitamin D supplementation especially during summer, which might negatively impact on women’s and fetal health. Moreover, buying and taking supplements that are not needed should not be recommended. In light of the finding that especially women with low socio-economic reveal a risk of inadequate intake of Vitamin D and taking into consideration that supplements are often not paid by health insurance, this should not be supported.

On my comment on excluding vegetarians and vegans, the author’s response was “In our study population there were only 9 women declaring vegetarian or vegan diet and the small sample size was the only reason why they were excluded from the analysis (these women can differ from the rest of the population not only in the type of food consumed but also in terms of other lifestyle-related characteristics such as recreational physical activity level).”

This would have actually been a novel and interesting point. Here, the authors argument that diet and lifestyle are tightly associated. However, they discuss their own finding that low socio-economic status is correlated to poor nutrition as a novelty. I agree that the number of subjects is low, but this would have actually been of interest, while most of the other correlations found in this manuscript are not.

Author Response

Dear Reviewer,

Thank you all for all your work on our manuscript “Determinants of an Adequate Essential Elements and Vitamins Status during Pregnancy”.

We truly appreciate and respect the comments stated by you in the second round of the review.

Below you can find responses to each of your specific comments.

Best regards,

Kinga Polanska
